# Genetic evidence that Chinese chestnut cultivars in Japan are derived from two divergent genetic structures that originated in China

**Sogo Nishio** [1]*, **Shuan Ruan**[2], **Yutaka Sawamura**[1], **Shingo Terakami**[1], **Norio Takada**[1], **Yukie Takeuchi**[1], **Toshihiro Saito**[1], **Eiich Inoue**[3]

**1** Institute of Fruit Tree and Tea Science, NARO, Tsukuba, Ibaraki, Japan, **2** Liaoning Institute of Economic Forestry, Dalian, Liaoning, China, **3** College of Agriculture, Ibaraki University, Ami-machi, Ibaraki, Japan

* nishios@affrc.go.jp

**Data Availability Statement:** All relevant data are within the manuscript and its Supporting Information files.

## Abstract

The Chinese chestnut (*Castanea mollissima* Bl.) was introduced into Japan about 100 years ago. Since then, a number of Chinese chestnut cultivars and Japanese–Chinese hybrid cultivars have been selected by farmers and plant breeders, but little information has been available about their origins and genetic relationships. A classification based on simple sequence repeat markers was conducted using 230 cultivars including Japanese chestnut (*Castanea crenata* Sieb. et Zucc.) cultivars originated in Japan, Japanese–Chinese hybrid cultivars, and Chinese chestnut cultivars originated in both Japan and China. First, a search for synonyms (cultivars with identical genotypes) revealed 23 synonym groups among the Chinese chestnut cultivars, and all but one cultivar from each synonym group was omitted from further analyses. Second, genetic structure analysis showed a clear division between Japanese and Chinese chestnut, and most of the Japanese and Chinese cultivars had a simple genetic structure corresponding to the expected species. On the other hand, most Japanese–Chinese hybrid cultivars had admixed genetic structure. Through a combination of parentage and chloroplast haplotype analyses, 16 of the 18 hybrid cultivars in this study were inferred to have parent–offspring relationships with other cultivars originated in Japan. Finally, Bayesian clustering and chloroplast haplotype analysis showed that the 116 Chinese chestnut cultivars could be divided into two groups: one originated in the Hebei region of China and the other originated in the Jiangsu and Anhui regions of China. The Chinese chestnut cultivars selected in Japan showed various patterns of genetic structure including Hebei origin, Jiangsu or Anhui origin, and admixed. The chestnut cultivar genetic classifications obtained in this study will be useful for both Japanese and Chinese chestnut breeding programs.

**Funding:** The author(s) received no specific funding for this work.

**Competing interests:** The authors have declared that no competing interests exist.

## Introduction

Japanese chestnut (*Castanea crenata* Sieb. et Zucc.) and Chinese chestnut (*C. mollissima* Bl.) are naturally distributed throughout Japan and China, respectively. Both species are reported to have been prehistorically domesticated [1–3] and are still economically important for production of edible nuts. These two species have large differences in both genetic and morphological properties. Japanese chestnut has large fruit size, adherent pellicle, and comparatively good yield, whereas Chinese chestnut has small nut size, easy-to-peel pellicle, and low yield [4]. In addition, the yellow brown shoot and pubescence at the nut tip are characters specific to Chinese chestnut and can be used to distinguish between the two species.

Interspecific hybridization has been used in chestnut breeding programs all over the world [4,5]. The breeding objectives of such programs have varied depending on the species. In the United States, the chestnut blight resistance gene from Chinese chestnut has been introduced into American chestnut (*C. dentata* [Marsh.] Borkh.) [6,7]. To improve European chestnut (*C. sativa* Mill.), quantitative trait loci for resistance to *Phytophthora cinnamomic* and chestnut gall wasp were introduced from Japanese chestnut [8,9]. In Japan, Chinese chestnut has been applied in Japanese chestnut breeding programs to improve nut quality and pellicle peelability [10,11]. Thus, it is important to clarify the genetic relationships among chestnut cultivars of different species.

According to Tanaka [12] and Isaki [13], Chinese chestnut was introduced into Japan about 100 years ago and attempts were made to cultivate it in Japan. However, this species was not suited to the Japanese climate [12] and had high susceptibility to the chestnut gall wasp [13]. Some local farmers in Japan planted nuts of Chinese chestnut and selected for adaptation to the local climate and conditions, leading to the development of Chinese chestnut cultivars such as 'Houji', 'Miyagawa', 'Hinoharu', and 'Aioi' in the early 20th century [13] Subsequently, the Japanese–Chinese hybrid cultivar 'Riheiguri' was selected by a local farmer and became one of the leading cultivars in Japan, accounting for 7% of total chestnut cultivation area in Japan in 2016. Even though this cultivar does not give high yields, its nut quality is highly valued by consumers and producers.

A number of Chinese chestnut and Japanese–Chinese hybrid cultivars have been collected and preserved at the NARO (National Agriculture and Food Organization) Genebank (www.gene.affrc.go.jp) and can be readily used for research and breeding purposes. However, information on the origin and genetic structure of these cultivars has been unclear or limited, since most of them were developed by local farmers and local agricultural experiment stations more than 50 years ago. Also, it is quite difficult to distinguish between pure Chinese chestnut and Japanese–Chinese hybrids by nut appearance. Although Japanese and Chinese chestnut readily produce interspecific hybrids, the extent of admixing from other species has not been clarified by molecular marker analyses.

The Chinese chestnut has the widest distribution among the chestnut species [14]. The northern range reaches 41˚N, following the ranges of the Yanshan Mountains, and the southern range extends to 18˚N on the Wuzhi Mountain of Hainan Island. Due to its large distribution, the Chinese chestnut is presumed to have higher genetic diversity than other chestnut species [15]. Several reports have suggested that central China, possibly the Shengnongia region near the Chang Jiang River, is the center of genetic diversity of Chinese chestnut and is one of the candidates for a refugium of this species [15–17]. On the other hand, Zhang and Liu [18] suggested that the southwest area of China is the center of diversity. So far, no obvious genetic structure corresponding to geographical location has been found by clustering analyses, suggesting that human-mediated transportation might have affected the wild chestnut population structure [17]. While a large number of chloroplast haplotypes were identified

within wild chestnut populations, only two haplotypes were identified in a collection of cultivars [19]. Ovesná et al. [20] also suggested that the genetic variability of Chinese chestnut cultivars was less than that of wild populations. Thus, several studies have examined the genetic diversity of wild chestnut and the genetic relationship between wild populations and cultivar collections. However, classifications based on clustering analyses using a large number of chestnut cultivars collected from diverse regions have not previously been conducted. In addition, the genetic relationship between Chinese chestnut cultivars within and outside of China has been unclear.

The classification and clustering of chestnut cultivars in Europe and Japan have been assessed using simple sequence repeats (SSRs) [21–25]. Because SSRs are highly reliable markers, they have been used to conduct Bayesian structure analyses and to identify synonyms (differently named cultivars with identical genotypes), homonyms (different genotypes with the same cultivar name), and parent–offspring relationships. The Bayesian clustering analyses usually correspond to region-based classification or prevalent nut use. On the other hand, humans have carried scions between different locations, resulting in some cultivars showing unexpected cultivar origin and population structure [25]. In fact, many synonym groups contain local cultivars from different regions [21,24] and parentage analyses have clarified that traditional cultivars have contributed to the appearance of many local cultivars [23,25].

The main objective of the present study was to clarify the origin and genetic characteristics of Chinese chestnut cultivars in Japan. We used a three-step strategy for cultivar classification. The first step was to identify synonym groups and eliminate duplicate genotypes prior to further analyses. The second was to clarify the genetic relationship between Japanese–Chinese hybrids and Chinese chestnut cultivars in Japan by using highly reliable materials as references, i.e., Japanese and Chinese chestnut cultivars originated in Japan and China, respectively. The third was to clarify the genetic relationship between Chinese chestnut cultivars selected in Japan and those originated in China. Identification of synonyms, parentage, and genetic relationships among cultivars would be useful for chestnut breeding programs and further genetic classification studies.

## Materials and methods

### Plant materials and DNA extraction

The 12 groups (230 cultivars) used in this study are shown in Tables 1 and S1. The Japanese chestnut, Japanese–Chinese hybrid, and Chinese chestnut cultivars from Japan are preserved at the NARO Genebank (www.gene.affrc.go.jp). These include Japanese chestnut local cultivars that originated in the Kanto region of Japan (designated J_KA), the Tanba region of Japan (J_TA), other regions of Japan (J_OJ), and Korea (KOR); Japanese–Chinese chestnut hybrids developed in Japan (HYB); and Chinese chestnut cultivars selected in Japan (C_SJ) and introduced from other countries (C_IO). The Chinese chestnut cultivars originated in China were provided by the Liaoning Province Economic Forest Research Institute, Shandong Institute of Pomology, and Hebei Agriculture and Forestry Academy of Sciences Changli Guoshu Institute. Chinese chestnut cultivar groups in China were defined by region of origin, i.e., Hebei (C_HE), Shandong (C_SH), Anhui (C_AN), Jiangsu (C_JI), and other regions of China (C_OR). To avoid using duplicate genotypes in the analysis, only one cultivar was used from each synonym group identified by Nishio et al. [24]. Genomic DNA was extracted from young leaves with a DNeasy Plant Mini Kit (Qiagen, Germany) according to the manufacturer's instructions.

**Table 1. Names, accession numbers, and genotype information for the 230 cultivars used in this study.**

| Genotype | Cultivar | Origin | Code (group number) |
|---|---|---|---|
| 1 | Arima | Kanto (Japan) | J_KA (1) |
| 2 | ChuutanA | Kanto (Japan) | J_KA (1) |
| 3 | Gosha | Kanto (Japan) | J_KA (1) |
| 4 | Hassaku | Kanto (Japan) | J_KA (1) |
| 5 | Moriwase | Kanto (Japan) | J_KA (1) |
| 6 | Nakatetanba | Kanto (Japan) | J_KA (1) |
| 7 | Odai | Kanto (Japan) | J_KA (1) |
| 8 | Osaya | Kanto (Japan) | J_KA (1) |
| 9 | Senri | Kanto (Japan) | J_KA (1) |
| 10 | Shichifukuwase | Kanto (Japan) | J_KA (1) |
| 11 | Taishouwase | Kanto (Japan) | J_KA (1) |
| 12 | Toyotamawase | Kanto (Japan) | J_KA (1) |
| 13 | Tsunehisa | Kanto (Japan) | J_KA (1) |
| 14 | Yamatowase | Kanto (Japan) | J_KA (1) |
| 15 | Choubei | Tanba (Japan) | J_TA (2) |
| 16 | Choukouji | Tanba (Japan) | J_TA (2) |
| 17 | Daihachi | Tanba (Japan) | J_TA (2) |
| 18 | Fukunami | Tanba (Japan) | J_TA (2) |
| 19 | Fukunishi | Tanba (Japan) | J_TA (2) |
| 20 | Ginyose | Tanba (Japan) | J_TA (2) |
| 21 | Higan | Tanba (Japan) | J_TA (2) |
| 22 | Ichiemon | Tanba (Japan) | J_TA (2) |
| 23 | Imakita | Tanba (Japan) | J_TA (2) |
| 24 | Kanotsume | Tanba (Japan) | J_TA (2) |
| 25 | Kenagaginyose | Tanba (Japan) | J_TA (2) |
| 26 | Kinseki | Tanba (Japan) | J_TA (2) |
| 27 | Kinyoshi | Tanba (Japan) | J_TA (2) |
| 28 | Konishiki | Tanba (Japan) | J_TA (2) |
| 29 | Matabei | Tanba (Japan) | J_TA (2) |
| 30 | Ogawa teteuchi | Tanba (Japan) | J_TA (2) |
| 31 | Otomune | Tanba (Japan) | J_TA (2) |
| 32 | Shimokatsugi | Tanba (Japan) | J_TA (2) |
| 33 | Shuuhouwase | Tanba (Japan) | J_TA (2) |
| 34 | Tajiriginyose | Tanba (Japan) | J_TA (2) |
| 35 | Yakko | Tanba (Japan) | J_TA (2) |
| 36 | Akachiu | Other regions in Japan | J_OJ (3) |
| 37 | Banseki | Other regions in Japan | J_OJ (3) |
| 38 | Buzen | Other regions in Japan | J_OJ (3) |
| 39 | Dengorou | Other regions in Japan | J_OJ (3) |
| 40 | Ganne | Other regions in Japan | J_OJ (3) |
| 41 | Hataya oguri | Other regions in Japan | J_OJ (3) |
| 42 | Ichikawawase | Other regions in Japan | J_OJ (3) |
| 43 | Kasaharawase | Other regions in Japan | J_OJ (3) |
| 44 | Katayama | Other regions in Japan | J_OJ (3) |
| 45 | Kinshuu | Other regions in Japan | J_OJ (3) |
| 46 | Ninomiya | Other regions in Japan | J_OJ (3) |
| 47 | Obiwase | Other regions in Japan | J_OJ (3) |

(*Continued*)

**Table 1.** (Continued)

| Genotype | Cultivar | Origin | Code (group number) |
|---|---|---|---|
| 48 | Obuse 3 | Other regions in Japan | J_OJ (3) |
| 49 | Okoma | Other regions in Japan | J_OJ (3) |
| 50 | Ooharaguri | Other regions in Japan | J_OJ (3) |
| 51 | Saimyouji 1 | Other regions in Japan | J_OJ (3) |
| 52 | Tanabata | Other regions in Japan | J_OJ (3) |
| 53 | Tanoue 1 | Other regions in Japan | J_OJ (3) |
| 54 | Terai | Other regions in Japan | J_OJ (3) |
| 55 | Togenashi | Other regions in Japan | J_OJ (3) |
| 56 | Tsuchidawase | Other regions in Japan | J_OJ (3) |
| 57 | Waseginzen | Other regions in Japan | J_OJ (3) |
| 58 | Yamaguchiwase | Other regions in Japan | J_OJ (3) |
| 59 | Yamaguchiwase 2 | Other regions in Japan | J_OJ (3) |
| 60 | Yourou | Other regions in Japan | J_OJ (3) |
| 61 | Buyu 3 | Korea | KOR (4) |
| 62 | Hamjung 3 | Korea | KOR (4) |
| 63 | Jungbu 26 | Korea | KOR (4) |
| 64 | Jungbu 8 | Korea | KOR (4) |
| 65 | Jungbu 9 | Korea | KOR (4) |
| 66 | Pochun B-1 | Korea | KOR (4) |
| 67 | Ikaba | Hyogo | HYB (5) |
| 68 | Hayashi 1 | Gifu | HYB (5) |
| 69 | Hayashi 3 | Gifu | HYB (5) |
| 70 | Hayashi amaguri | Gifu | HYB (5) |
| 71 | Hyogo 493 | Hyogo | HYB (5) |
| 72 | Kurakata amaguri | Tokyo | HYB (5) |
| 73 | Nishiharima | Hyogo | HYB (5) |
| 74 | Omatsuguri | Ehime | HYB (5) |
| 75 | Riheiguri | Gifu | HYB (5) |
| 76 | Senshu amaguri | Akita | HYB (5) |
| 77 | Shimaki 1 | Ibaraki | HYB (5) |
| 78 | Shimaki 2 | Ibaraki | HYB (5) |
| 79 | Shimaki 3 | Ibaraki | HYB (5) |
| 80 | Shimaki 4 | Ibaraki | HYB (5) |
| 81 | Shimaki 5 | Ibaraki | HYB (5) |
| 82 | Shimaki 6 | Ibaraki | HYB (5) |
| 83 | Wasetenshin | Unknown | HYB (5) |
| 84 | Yamewase | Fukuoka | HYB (5) |
| 85 | Aioi | Aichi | C_SJ (6) |
| 86 | C-4 | Kanagawa | C_SJ (6) |
| 87 | Gifu 1 | Gifu | C_SJ (6) |
| 88 | Hakuri | Nagano | C_SJ (6) |
| 89 | Hinoharu | Yamanashi | C_SJ (6) |
| 90 | Hinoharu 2 | Yamanashi | C_SJ (6) |
| 91 | Houji 354 | Kochi | C_SJ (6) |
| 92 | Houji 445 | Kochi | C_SJ (6) |
| 93 | Houji 350 | Kochi | C_SJ (6) |
| 94 | Houji 446 | Kochi | C_SJ (6) |

(*Continued*)

**Table 1.** (Continued)

| Genotype | Cultivar | Origin | Code (group number) |
|---|---|---|---|
| 95 | Houji 480 | Kochi | C_SJ (6) |
| 96 | Hyogo shinaguri | Hyogo | C_SJ (6) |
| 97 | Iwate amaguri | Unknown | C_SJ (6) |
| 98 | Kahoku 10 | Tsukuba | C_SJ (6) |
| 99 | Kanan 56 | Tsukuba | C_SJ (6) |
| 100 | Konan 22 | Tsukuba | C_SJ (6) |
| 101 | Konan 36 | Tsukuba | C_SJ (6) |
| 102 | Konan 52 | Tsukuba | C_SJ (6) |
| 103 | Kousei 2 | Tsukuba | C_SJ (6) |
| 104 | Miyagawa 100 | Yamanashi | C_SJ (6) |
| 105 | Miyagawa 18 | Yamanashi | C_SJ (6) |
| 106 | Miyagawa 84 | Yamanashi | C_SJ (6) |
| 107 | Miyagawa 85 | Yamanashi | C_SJ (6) |
| 108 | Miyagiguri | Unknown | C_SJ (6) |
| 109 | Houji 360 | Kochi | C_SJ (6) |
| 110 | Tsuchida Amaguri | Gifu | C_SJ (6) |
| 111 | Yunba2 | Yamanashi | C_SJ (6) |
| 112 | Connecticut Yankee | U.S.A. | C_IO (7) |
| 113 | Hamden | U.S.A. | C_IO (7) |
| 114 | Nepal chestnut | Nepal | C_IO (7) |
| 115 | Sleeping Giant | U.S.A. | C_IO (7) |
| 116 | Tokuganri A | North Korea | C_IO (7) |
| 117 | 2399 | Hebei (China) | C_HE (8) |
| 118 | Dabanhong | Hebei (China) | C_HE (8) |
| 119 | Donglingmingzhu | Hebei (China) | C_HE (8) |
|  | Xigou 7 | Hebei (China) | C_HE (8) |
| 120 | Guanting 10 | Hebei (China) | C_HE (8) |
| 121 | Houhanzhuang 20 | Hebei (China) | C_HE (8) |
| 122 | Qiananli | Hebei (China) | C_HE (8) |
| 123 | Yanchangli | Hebei (China) | C_HE (8) |
| 124 | Yanfeng | Hebei (China) | C_HE (8) |
| 125 | Yankui | Hebei (China) | C_HE (8) |
|  | Yancheng 3 | Unknown (China) | C_OR (12) |
| 126 | Yanshanduanzhi | Hebei (China) | C_HE (8) |
|  | Dahongpao | Unknown (China) | C_OR (12) |
|  | Laiyangduanzhi | Shandong (China) | C_SH (9) |
|  | Laizhouduanzhi | Shandong (China) | C_SH (9) |
|  | Shuheduanzhi | Shandong (China) | C_SH (9) |
| 127 | Yanshanzaofeng | Hebei (China) | C_HE (8) |
|  | Xinzhuang 2 | Beijing (China) | C_OR (12) |
| 128 | Zaofeng | Hebei (China) | C_HE (8) |
| 129 | Zundali | Hebei (China) | C_HE (8) |
| 130 | Chuixhili 2 | Shandong (China) | C_SH (9) |
| 131 | Fulaiwuhuali | Shandong (China) | C_SH (9) |
| 132 | Haifeng | Shandong (China) | C_SH (9) |
| 133 | Hongguang-LPEFRI | Shandong (China) | C_SH (9) |
| 134 | Hongli 1 | Shandong (China) | C_SH (9) |

(*Continued*)

**Table 1.** (Continued)

| Genotype | Cultivar | Origin | Code (group number) |
|---|---|---|---|
| 135 | Hongli 3 | Shandong (China) | C_SH (9) |
| | Hongli | Unknown (China) | C_OR (12) |
| 136 | Huafeng | Shandong (China) | C_SH (9) |
| | Taianboke-HAAFS | Unknown (China) | C_OR (12) |
| 137 | Huagai | Shandong (China) | C_SH (9) |
| 138 | Huaguang | Shandong (China) | C_SH (9) |
| 139 | Jinfeng | Shandong (China) | C_SH (9) |
| | Hongguang-HAAFS | Shandong (China) | C_SH (9) |
| | Lianxujieguo | Unknown (China) | C_OR (12) |
| 140 | Junandagongshu | Shandong (China) | C_SH (9) |
| 141 | Mengshankuili | Shandong (China) | C_SH (9) |
| 142 | Shandongchuizhi 3 | Shandong (China) | C_SH (9) |
| 143 | Shandongchushuhong | Shandong (China) | C_SH (9) |
| | Shimenzaoshuo | Shandong (China) | C_SH (9) |
| 144 | Shifeng | Shandong (China) | C_SH (9) |
| | Huaifeng | Unknown (China) | C_OR (12) |
| 145 | Songjiazao-HAAFS | Shandong (China) | C_SH (9) |
| | Wuhua | Shandong (China) | C_SH (9) |
| 146 | Songjiazao-SIP | Shandong (China) | C_SH (9) |
| | Guangxiyouli | Unknown (China) | C_OR (12) |
| 147 | Taianaisheng | Shandong (China) | C_SH (9) |
| 148 | Taianboke-SIP | Shandong (China) | C_SH (9) |
| 149 | Tancheng 207 | Shandong (China) | C_SH (9) |
| 150 | Tanchengyouguangli | Shandong (China) | C_SH (9) |
| 151 | Weifeng | Shandong (China) | C_SH (9) |
| 152 | Yanming | Shandong (China) | C_SH (9) |
| 153 | Yimengduanzhi | Shandong (China) | C_SH (9) |
| | Duanzhiboke | Unknown (China) | C_OR (12) |
| 154 | Yimengkuili | Shandong (China) | C_SH (9) |
| 155 | Chali | Jiangsu (China) | C_JI (10) |
| 156 | Chongyangpu | Jiangsu (China) | C_JI (10) |
| 157 | Dadiqing | Jiangsu (China) | C_JI (10) |
| | Shuhe 10 | Shandong (China) | C_SH (9) |
| | Shuhe 11 | Shandong (China) | C_SH (9) |
| | Shuhe 14 | Shandong (China) | C_SH (9) |
| 158 | Duanmaojiaozha | Jiangsu (China) | C_JI (10) |
| 159 | Guihuali | Jiangsu (China) | C_JI (10) |
| 160 | Jiujiazhong | Jiangsu (China) | C_JI (10) |
| 161 | Paoche 7 | Jiangsu (China) | C_JI (10) |
| 162 | Qingmaoruanci-HAAFS | Jiangsu (China) | C_JI (10) |
| 163 | Qingzha | Jiangsu (China) | C_JI (10) |
| 164 | Yixingdahongpao | Jiangsu (China) | C_JI (10) |
| 165 | Ershuizao | Anhui (China) | C_AN (11) |
| 166 | Hefeichushuhong | Anhui (China) | C_AN (11) |
| | Shuangjili | Jiangxi (China) | C_OR (12) |
| 167 | Mifengqiu | Anhui (China) | C_AN (11) |
| | Paoche 2 | Jiangxi (China) | C_JI (10) |

(*Continued*)

**Table 1.** (Continued)

| Genotype | Cultivar | Origin | Code (group number) |
|---|---|---|---|
| 168 | Niandiban | Anhui (China) | C_AN (11) |
| 169 | Shuanghedahongpao | Anhui (China) | C_AN (11) |
| | Tedazaoyou | Unknown (China) | C_OR (12) |
| 170 | Shuchengdahongpao | Anhui (China) | C_AN (11) |
| | Liyang 1302 | Fujian (China) | C_OR (12) |
| | Qingmaoruanci-SIP | Jiangsu (China) | C_JI (10) |
| 171 | Shuizao 2–11 | Anhui (China) | C_AN (11) |
| 172 | Yebianza | Anhui (China) | C_AN (11) |
| 173 | Yelicang | Anhui (China) | C_AN (11) |
| | Taili 1 | Shandong (China) | C_SH (9) |
| 174 | 214 | Unknown (China) | C_OR (12) |
| 175 | Duanzhatou | Unknown (China) | C_OR (12) |
| 176 | Guangxili | Unknown (China) | C_OR (12) |
| | Zajiao 35 | Unknown (China) | C_OR (12) |
| 177 | Hongyouli | Guangxi (China) | C_OR (12) |
| 178 | Houzhuang 2 | Unknown (China) | C_OR (12) |
| 179 | Huaiduanhua | Unknown (China) | C_OR (12) |
| | Huaiwuhua | Unknown (China) | C_OR (12) |
| 180 | Huaihuang | Beijing (China) | C_OR (12) |
| 181 | Huaiyanhong | Unknown (China) | C_OR (12) |
| 182 | Huangpeng | Unknown (China) | C_OR (12) |
| 183 | Jiandingyouli | Unknown (China) | C_OR (12) |
| 184 | Jinpingduanchui | Jiangxi (China) | C_OR (12) |
| 185 | Juhong | Unknown (China) | C_OR (12) |
| | Zhongming | Unknown (China) | C_OR (12) |
| 186 | Kui 1–3 | Unknown (China) | C_OR (12) |
| 187 | Kuili | Zhejiang (China) | C_OR (12) |
| | Duancidaqing | Unknown (China) | C_OR (12) |
| 188 | Laokuili | Unknown (China) | C_OR (12) |
| 189 | Linfeng | Unknown (China) | C_OR (12) |
| 190 | Luotianzaoli | Hubei (China) | C_OR (12) |
| 191 | Panzhuang 1 | Unknown (China) | C_OR (12) |
| 192 | Qianci | Hubei (China) | C_OR (12) |
| 193 | Qiannanyu 3 | Unknown (China) | C_OR (12) |
| 194 | Shandongwuming | Unknown (China) | C_OR (12) |
| 195 | Shangguang | Zhejiang (China) | C_OR (12) |
| 196 | Taiboke | Unknown (China) | C_OR (12) |
| 197 | Tali | Hunan (China) | C_OR (12) |
| 198 | Yanchang | Beijing (China) | C_OR (12) |
| | Yanhong | Beijing (China) | C_OR (12) |
| 199 | Yebanli | Unknown (China) | C_OR (12) |
| 200 | Yinxuan 3 | Guangxi (China) | C_OR (12) |

## Genetic markers

The 230 cultivars were genotyped for 31 nuclear SSRs [26–28] (S2 Table) and 5 chloroplast SSRs (cpSSRs) (Cmcs1–3, Cmcs5, and Cmcs7) [29]. PCR amplification was performed in a 10-

μL solution containing 5 μL of 2× Green GoTaq G2 Hot Start Master Mix (0.4 mM each dNTP, Taq DNA polymerase, and 4 mM MgCl$_2$, pH 8.5; Promega, Madison, WI, USA), 20 pmol of each forward primer labeled with a fluorescent dye (5-FAM or 5-HEX) and unlabeled reverse primer, and 2.5 ng of genomic DNA. Amplification was performed in 35 cycles of 94˚C for 1 min, 55˚C for 1 min, and 72˚C for 2 min. PCR products were separated and detected with a 3130xl Genetic Analyzer (Life Technologies, Carlsbad, CA, USA). The size of each amplified band was determined by comparison with a set of internal-standard DNA fragments (400HD ROX, Life Technologies) in GeneMapper software v. 5.0 (Life Technologies).

## Data analysis

Prior to analysis of population structure and parent–offspring relationships, synonym groups were identified using the 31 nuclear SSR markers to analyze the 230 cultivars. After eliminating duplicate genotypes, a set of 200 unique cultivars (see Results) was used for further analyses. The probability of identity (PI) for each locus and for the whole SSR set (Cumulative PI) was calculated using the software Gimlet v1.3.3 [30] to check the power of discrimination. Chloroplast haplotypes were determined by using the 5 cpSSRs (S1 Table). Cultivars that had the identical combination of alleles for all 5 cpSSRs were considered to carry the same haplotype. The positions of the 5 cpSSRs within the complete *Castanea mollissima* chloroplast genome are shown in S1 Table.

Bayesian statistical inference on the population structure was performed by using Structure 2.3.4 software [31] with the independent model for allele frequency, without any prior information about the origin of each cultivar. First, the 200 cultivars representing unique genotypes were used to clarify the genetic structure of Japanese and Chinese chestnut and their hybrids. Next, only the pure Chinese chestnut cultivars were used in a second structure analysis to examine genetic structures within that species. The analysis was run 10 times for each value of *K* (number of inferred ancestral populations) from 2 to 8 for 1,000,000 iterations after a burn-in period of 100,000 iterations. Evanno et al.'s [32] criterion of Δ*K* was used to estimate the appropriate *K* value. Principal coordinate analysis (PCoA) was performed in GenAlEx 6.5 from the pairwise genetic distances obtained with the covariance-standardized method [33].

For the Japanese–Chinese hybrid cultivar group, putative parent–offspring relationships were calculated with the parent calculation program MARCO [34], which identifies possible parents from among the genotypes in a set of cultivars. Genotypes were considered to have a parent–offspring relationship if they shared at least one allele per SSR locus, with the exception that a discrepancy at a single SSR locus was accepted to allow for possible genotyping errors, presence of null alleles, or mutation, as previously proposed [25,35–38]. To determine whether a parent was the seed parent or pollen parent, chloroplast haplotype data were used.

For the Chinese chestnut groups that had more than 8 cultivars each (C_SJ, C_HE, C_SH, C_AN, C_JI, and C_OR), the observed heterozygosity ($H_O$), expected heterozygosity ($H_E$), and inbreeding coefficient (F) were calculated using GenAlEx v. 6.5 software [33], and allelic richness (AR, $n = 9$) was calculated using the R package Hierfstat [39].

## Evaluation of phenotypic traits

Nut harvest date and nut weight were recorded in 2001–2003 for cultivars preserved at the NARO Genebank, which include Japanese chestnut cultivars, Japanese–Chinese hybrid cultivars, and Chinese chestnut cultivars preserved in Japan (S3 Table). To evaluate nut harvest date and nut weight, each bur was harvested when it had changed from green to brown and had begun to split open or had dropped. The nuts were removed from the burs and the number of nuts harvested on a given day was recorded for each tree. Burs and nuts were harvested

every three or four days from late August to October. The harvest date for each nut was expressed as the number of days after July 31 (i.e., August 1 = day 1), and the average value of nut harvest date for each tree was used as its score for this trait. Nut weight (g) per nut was measured on a digital scale on each harvest date. The average nut weight was calculated as the total nut weight divided by the total number of intact nuts.

For Chinese chestnut cultivars originated in China, nut harvest date and nut weight could not be obtained because only DNA samples were available.

## Results

### Identification of synonym groups

We could differentiate all but 53 of the 230 chestnut cultivars with the 31 nuclear SSR markers. These 53 cultivars were divided into 23 synonym groups, each consisting of 2 to 5 cultivars having the same genotypes at all SSR loci (Table 2). Consequently, we identified 200 unique genotypes from the 230 cultivars. The probability of identity (PI) for each locus ranged from 0.016 for PRG79 to 0.269 for PRD83 (mean = 0.099), whereas the total PI was $6.56 \times 10^{-35}$. Synonym groups were found only within Chinese chestnut cultivars originated in China: none of the synonym groups contained cultivars originated in Japan. Some of the cultivars showing identical genotypes originated in different regions of China; however, most of them originated in regions adjacent to one another. Examples of synonym groups include 'Yanshanduanzhi' from Hebei and 'Laiyangduanzhi', 'Laizhouduanzhi', and 'Shuheduanzhi' from Shandong (Syn-10); 'Yanshanzaofeng' from Hebei and 'Xinzhuang 2' from Beijing (Syn-6); 'Mifengqiu'

**Table 2. Chinese chestnut cultivar groups with identical SSR genotypes for all 31 SSR markers.**

| Group | Cultivars | | | | |
|---|---|---|---|---|---|
| Syn-1 | Hefeichushuhong | Shuangjili | | | |
| Syn-2 | Mifengqiu | Paoche 2 | | | |
| Syn-3 | Shuanghedahongpao | Tedazaoyou | | | |
| Syn-4 | Shuchengdahongpao | Qingmaoruanci-SIP | Liyang 1302 | | |
| Syn-5 | Yelicang | Taili 1 | | | |
| Syn-6 | Yanshanzaofeng | Xinzhuang 2 | | | |
| Syn-7 | Yanchang | Yanhong | | | |
| Syn-8 | Donglingmingzhu | Xigou 7 | | | |
| Syn-9 | Yankui | Yancheng 3 | | | |
| Syn-10 | Yanshanduanzhi | Laiyangduanzhi | Laizhouduanzhi | Shuheduanzhi | Dahongpao |
| Syn-11 | Dadiqing | Shuhe 10 | Shuhe 11 | Shuhe 14 | |
| Syn-12 | Hongguang-HAAFS | Jinfeng | Lianxujieguo | | |
| Syn-13 | Hongli 3 | Hongli | | | |
| Syn-14 | Huafeng | Taianboke-HAAFS | | | |
| Syn-15 | Shifeng | Huaifeng | | | |
| Syn-16 | Songjiazao-HAAFS | Wuhua | | | |
| Syn-17 | Songjiazao-SIP | Guangxiyouli | | | |
| Syn-18 | Yimengduanzhi | Duanzhiboke | | | |
| Syn-19 | Duancidaqing | Kuili | | | |
| Syn-20 | Guangxili | Zajiao 35 | | | |
| Syn-21 | Huaiduanhua | Huaiwuhua | | | |
| Syn-22 | Juhong | Zhongming | | | |
| Syn-23 | Shandongchushuhong | Shimenzaoshuo | | | |

from Anhui and 'Paoche 2' from Jiangsu (Syn-2); and 'Hefeichushuhong' from Jiangsu and 'Shuangjili' from Anhui (Syn-1).

## Chloroplast haplotype frequency

In total, 6 chloroplast haplotypes were identified among the 200 unique cultivars by using 5 cpSSRs (Table 3). The three Japanese chestnut cultivar groups (J_KA, J_TA, and J_OJ) carried only HAP1, whereas the Chinese chestnut cultivar groups mainly carried HAP3 and HAP4. HAP2 was only found in the Korean cultivar 'Pochun B-1' (ID#66). HAP5 and HAP6 were only found in Chinese chestnut cultivars 'Kousei 2' (ID#103) and 'Duanzhatou' (ID#175), respectively. Among the Chinese chestnut cultivars, HAP3 was the only haplotype found in C_JI and C_AN (East central regions), whereas HAP4 dominated in C_HE (Northern region). The cultivars originated in Shandong (C_SH), which is located between Hebei and Jiangsu, carried both HAP3 and HAP4. Chinese chestnut cultivars from other regions of China (C_OR) and those from Japan (C_SJ) and other world areas (C_IO) also carried both HAP3 and HAP4.

## Genetic relationship between Japanese and Chinese chestnut cultivars and their hybrids

To clarify the genetic relationships between Japanese and Chinese chestnut cultivars and their hybrids, we performed Bayesian clustering analyses (Fig 1). The values of $\Delta(K)$ were much higher at $K = 2$ than at $K = 3$ to $K = 8$, so we constructed bar plot diagrams at $K = 2$. The "red" and "light green" shading indicate the Japanese and Chinese chestnut clusters, respectively. Most of the Japanese and Chinese chestnut cultivars had membership in a single cluster corresponding to their respective species. On the other hand, most Japanese–Chinese hybrid cultivars (HYB) had admixed structure, i.e., had approximately equal membership in both the "red" and "light green" clusters. Among the HYB cultivars, only 'Ikaba' (ID#67) had membership predominantly in the "light green" (Chinese) cluster. In both the Japanese and Chinese chestnut cultivar groups, a few cultivars had admixed structures (ratio of red:light green = 0.78:0.22 for 'Obiwase', 0.24:0.76 for 'Hamjung 3', and 0.16:0.84; for 'Miyagawa 18'). Some Chinese chestnut cultivars had very low membership in the "red" (Japanese) cluster, but the amount was too small to declare that they were hybrids (for example, red:light green = 0.04:0.96 for 'Tokuganri A' and 0.03:0.97 for '2399').

**Table 3. Haplotype frequencies in the set of 200 cultivars.**

|        | HAP1 | HAP2 | HAP3 | HAP4 | HAP5 | HAP6 |
|--------|------|------|------|------|------|------|
| J_KA   | 14   |      |      |      |      |      |
| J_TA   | 21   |      |      |      |      |      |
| J_OJ   | 25   |      |      |      |      |      |
| KOR    | 4    | 1    |      | 1    |      |      |
| HYB    | 6    |      | 10   | 2    |      |      |
| C_SJ   |      |      | 19   | 7    | 1    |      |
| C_IO   |      |      | 4    | 1    |      |      |
| C_HE   |      |      | 1    | 12   |      |      |
| C_SH   |      |      | 18   | 7    |      |      |
| C_JI   |      |      | 10   |      |      |      |
| C_AN   |      |      | 9    |      |      |      |
| C_OR   |      |      | 17   | 9    |      | 1    |

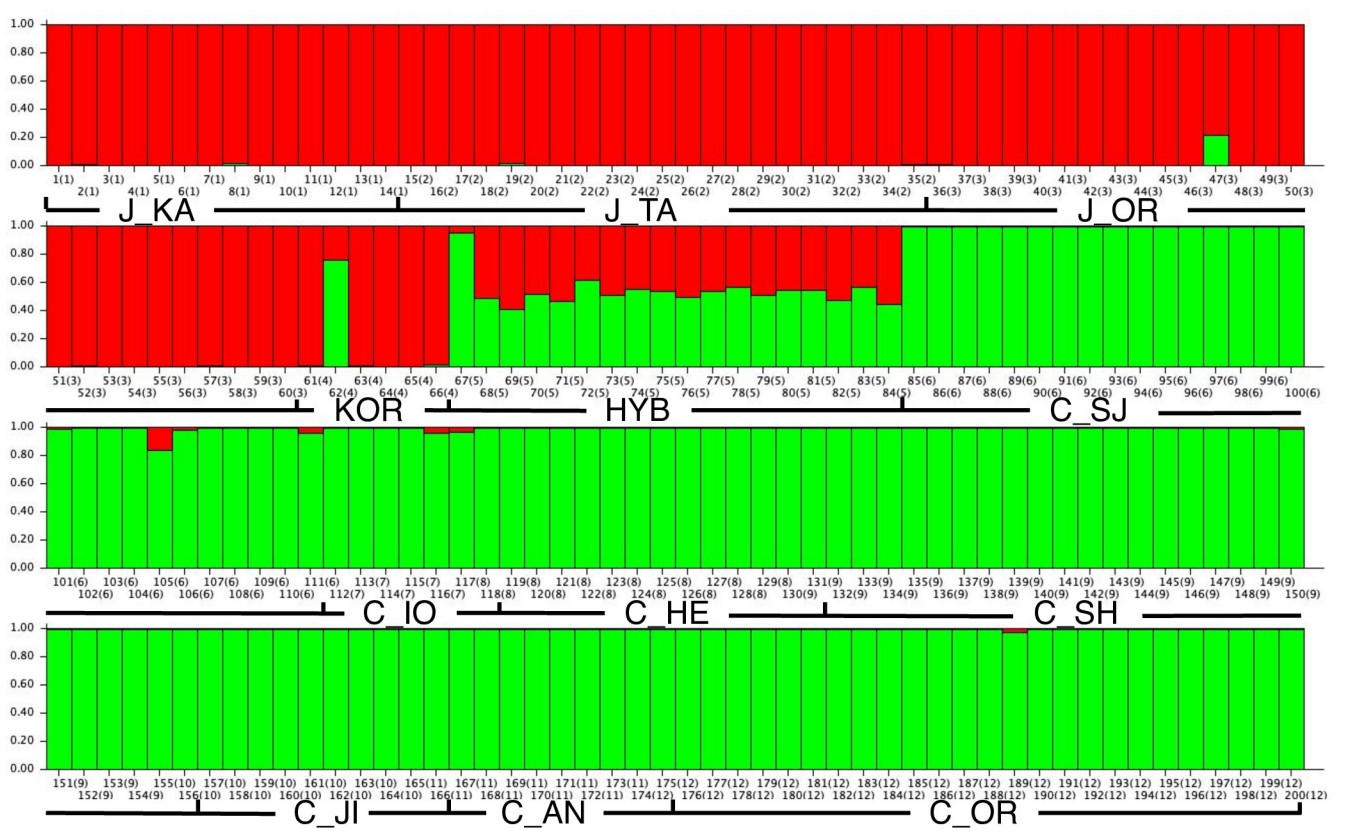

**Fig 1. Detailed bar plot diagram for $K = 2$ in the independent model using 66 Japanese chestnut cultivars, 18 Japanese–Chinese hybrids, and 116 Chinese chestnut cultivars.** The first number under each bar represents the individual accession ID number (1–200); the second number (in parentheses) represents the group number (1–12). ID numbers and groups are defined in Table 1.

### Putative parentage of Japanese–Chinese hybrid cultivars

Because some Japanese–Chinese hybrid cultivars had records indicating that they had been selected from crosses between Japanese and Chinese chestnut cultivars in Japan, we used MARCO software to perform parent–offspring relationship analysis using cultivars originated in Japan (Table 4). Out of the 18 Japanese–Chinese cultivars in this study, we could infer both parents for 6 cultivars ('Ikaba', 'Hayashi 1', 'Hayashi 3', 'Hayashi amaguri', 'Hyogo 493', and 'Yamewase'; ID#67–71, 84) and one parent for 10 cultivars ('Kurakata amaguri', 'Nishiharima', 'Omatsuguri', 'Senshu amaguri', 'Shimaki 1', 'Shimaki 2', 'Shimaki 3', 'Shimaki 4', 'Shimaki 5', and 'Shimaki 6'; ID#72–74, 76–82). Out of the six cultivars for which we were able to infer both parents, five cultivars were F1 hybrids between Japanese and Chinese chestnut and one cultivar, 'Ikaba' (ID#67), was presumed to be an offspring between Japanese–Chinese hybrid cultivar 'Riheiguri' (ID#75) and Chinese chestnut cultivar 'Gifu 1' (ID#87).

### Phenotypic trait evaluation of Japanese chestnut cultivars

Nut harvest date ranged from August 19 ['Hassaku' (ID#4)] to October 15 ['Shimokatsugi' (ID#32)] (average, September 18) for Japanese chestnut cultivars and from August 31 ['Hayashi 3' (ID#69)] to October 1 ['Shimaki 1' (ID#77)] (average, September 17) for Japanese–Chinese chestnut cultivars (S3 Table). The average nut harvest date of the Chinese chestnut cultivars preserved in Japan (September 29) was later than those of the Japanese and Japanese–

**Table 4. Putative parent–offspring relationships for Japanese–Chinese hybrid cultivars.**

| Offspring | Seed parent | Pollen parent |
|---|---|---|
| Ikaba | Riheiguri (H) | Gifu 1 (C) |
| Hayashi 1 | Houji 480 (C) | Kasaharawase (J) |
| Hayashi 3 | Houji 360 (C) | Kasaharawase (J) |
| Hayashi amaguri | Houji 360 (C) | Kanotsume (J) |
| Hyogo 493 | Kinseki (J) | Hyogo shinaguri (C) |
| Kurakata amaguri | Houji 480 (C) | Unknown |
| Nishiharima | Hyogo shinaguri (C) | Unknown |
| Omatsuguri | Unknown | Nakatetanba (J) |
| Senshu amaguri | Unknown | Tsuchidawase (J) |
| Shimaki 1 | Ganne (J) | Unknown |
| Shimaki 2 | Ganne (J) | Unknown |
| Shimaki 3 | Choubei (J) | Unknown |
| Shimaki 4 | Houji 446 (C) | Unknown |
| Shimaki 5 | Ganne (J) | Unknown |
| Shimaki 6 | Ganne (J) | Unknown |
| Yamewase | Houji 480 (C) | Nakatetanba (J) |
| Riheiguri | Unknown | Unknown |
| Wasetenshin | Unknown | Unknown |

"J", "C", and "H" in parentheses represent Japanese chestnut, Chinese chestnut and a hybrid, respectively.

Chinese chestnut hybrid cultivars and ranged from September 10 ['Kahoku 10 (ID#98) and "Miyagawa 84' (ID#106)] to October 17 ['Konan 22' (ID#100) and 'Kousei 2' (ID#103)]).

The average nut weight was large (24.7 g) in Japanese chestnut cultivars, intermediate (20.0 g) in Japanese–Chinese chestnut hybrids, and small (10.9 g) in Chinese chestnut cultivars preserved in Japan. The range was 11.9–40.9 g for Japanese chestnut cultivars, 11.9–29.5 g for Japanese–Chinese chestnut hybrids, and 5.2–17.6 g for Chinese chestnut cultivars in Japan. The average nut weight of Chinese chestnut cultivars that carried HAP3 (12.6 g) was larger than that of those carrying HAP4 (7.5g).

## Genetic relationships among Chinese chestnut cultivars in Japan and China

To determine the genetic relationships among Chinese chestnut cultivars from different areas, Bayesian clustering analysis was performed using only the 116 Chinese chestnut cultivars (Fig 2). The values of Δ(K) were highest at K = 2: values of Δ(K) at K = 3 to K = 8 were less than one-thousandth that at K = 2. The two Bayesian clusters strongly corresponded to the results of chloroplast haplotype analysis (Fig 3). Membership in the "red" cluster was dominant in cultivars from northern China (C_HE), all but one of which were HAP4, whereas membership in the "light green" cluster was dominant in cultivars from east central China (C_JI and C_AN), all of which were HAP3. The C_SJ, C_SH, and C_OR cultivars had membership in both the "red" and "light green" clusters. Although most of the C_SH cultivars were admixed, many of the C_SJ cultivars had membership in a single cluster (ID#92–98, 105–109, 111). In C_SJ, cultivars that carried HAP3 (S1 Table) belonged to the "light green" or admixed clusters (Fig 2; ID#85–95, 97, 99–102, 108–110), while most of the cultivars that carried HAP4 belonged to the "red" cluster (ID#98, 105–107, 111). To validate the results of the Bayesian clustering analysis, PCoA was conducted after classifying each cultivar as predominantly part of the "red" cluster (ratio of "red" > 0.8), the "light green" cluster (ratio of "light green" > 0.8), or admixed

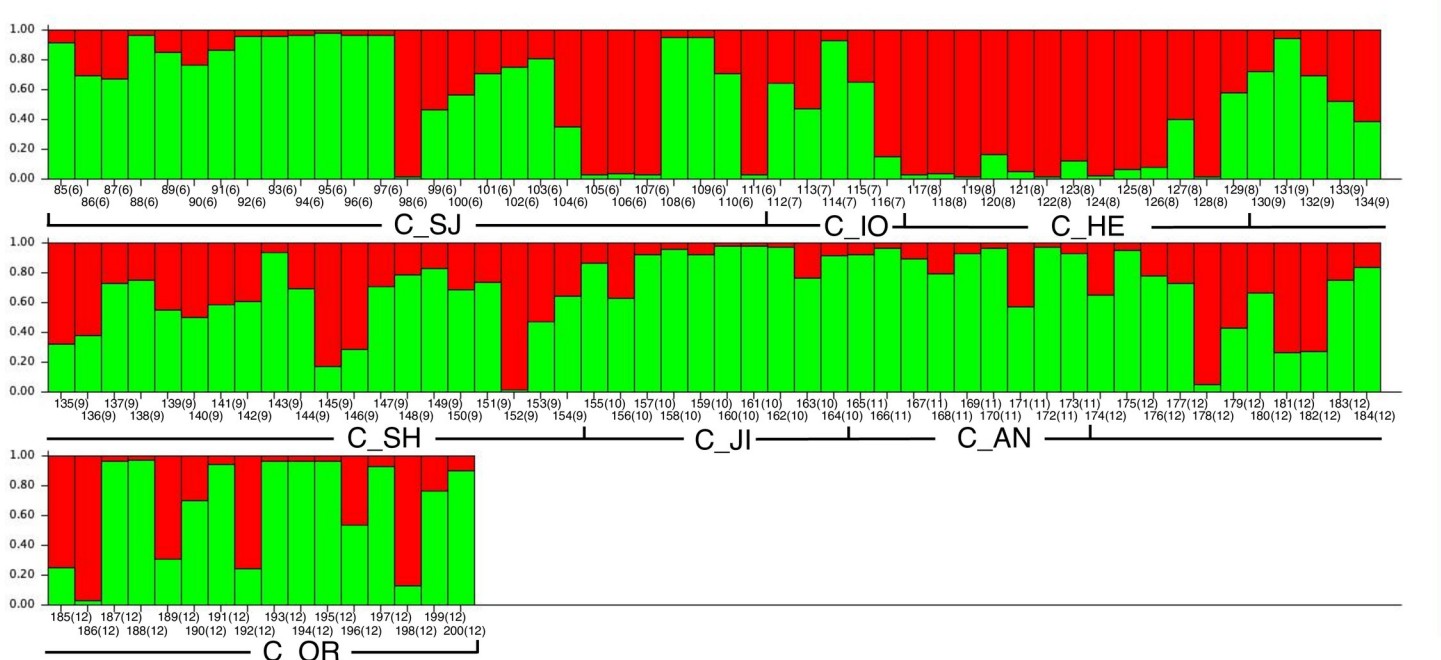

**Fig 2. Detailed bar plot diagram for _K_ = 2 in the independent model using 117 Chinese chestnut cultivars.** The first number under each bar represents the individual accession ID number (1–200); the second number (in parentheses) represents the group number (6–12). ID numbers and groups are defined in Table 1. Red and green are used to indicate each of the _K_ = 2 populations in this analysis and do not correspond to the same colors in Fig 1.

(0.8 > ratio of "red" > 0.2). The results of PCoA were similar to those of the Structure analysis, and the first informative PCo component corresponded to the separation between the "red" and "green" clusters (S1 Fig).

Genetic parameters were calculated to clarify the genetic diversity of Chinese chestnut cultivar groups that had more than 8 cultivars (Table 5). $H_O$ was the lowest for C_HE (0.457), whereas $H_O$ for C_AN (0.642) was slightly higher than for the other groups. $H_E$ and AR for C_HE were both lower than for the other groups. On the other hand, AR in C_SJ, C_SH, and C_OR, all of which had both HAP3 and HAP4 cultivars (Table 3), was higher than in the other groups. The inbreeding coefficient (_F_) was highest for C_SJ (0.085), whereas those for C_JI and C_AN were negative (−0.118 and −0.138, respectively).

## Discussion

Our set of 31 nuclear SSR loci proved to have an high discriminative power (total probability of identity: $6.56 \times 10^{-35}$) for the 200 unique cultivars. This value of the total probability of identity is quite low compared to those in other studies related to identification of synonyms ($3.73 \times 10^{-12}$–$2.99 \times 10^{-8}$) [40–42]. It is highly unlikely to detect false synonyms with the 31 nuclear SSR markers. We identified 23 synonym groups among the Chinese chestnut cultivars originated in China. A previous study using Japanese and Chinese chestnut cultivars in Japan had already identified two synonym groups for Chinese chestnut cultivars selected in Japan and one for Japanese–Chinese hybrid cultivars [24]. Synonyms have been commonly identified in chestnut cultivar collections [21–23]. One reason that chestnut cultivars may have many synonyms is that nut appearance is quite similar among cultivars. For most major fruit crops, fruit color would be a good characteristic to distinguish cultivars, but color differences are not helpful for distinguishing chestnut cultivars in most cases. Interestingly, synonyms

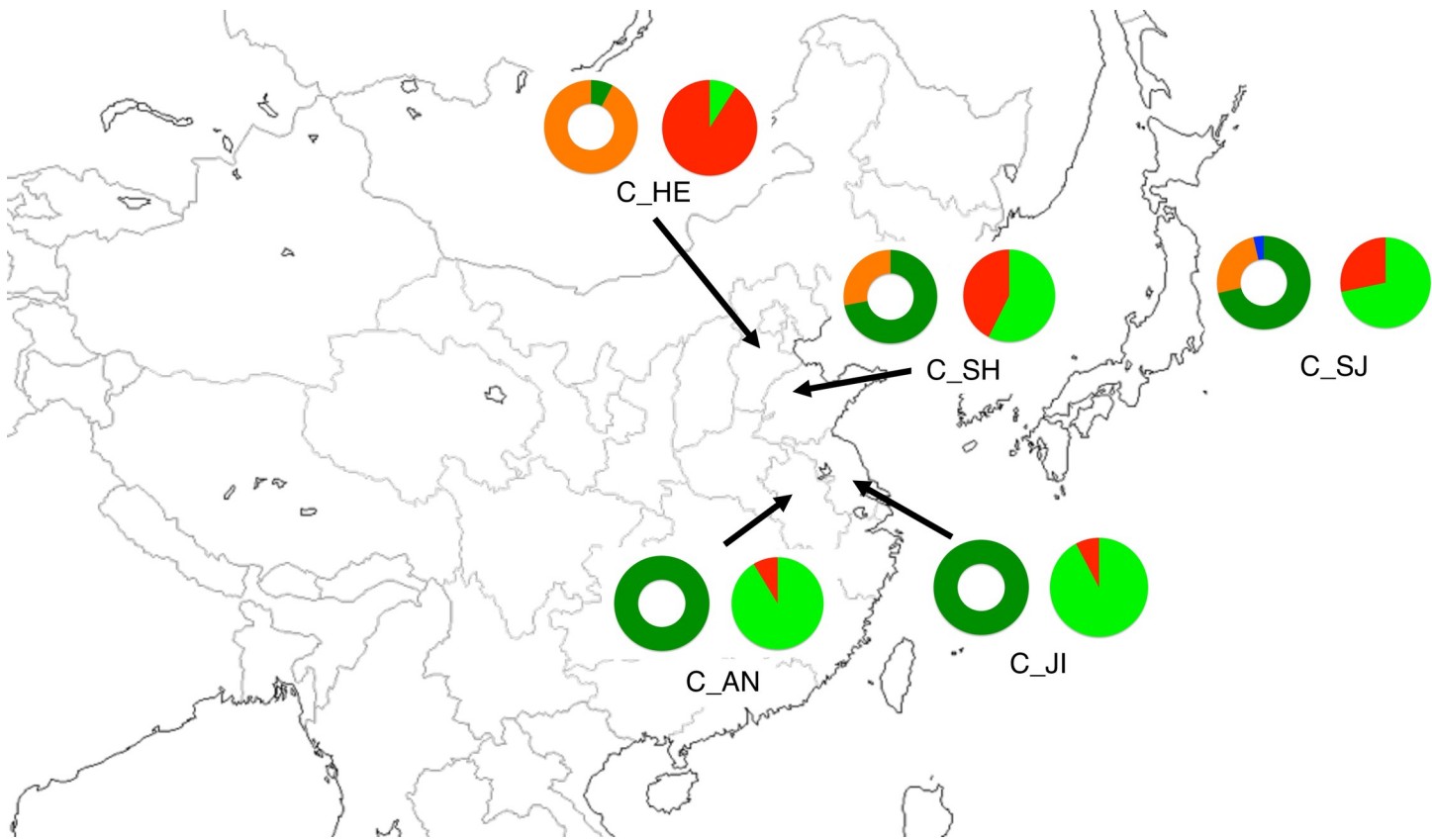

**Fig 3. Geographic locations of Chinese chestnut cultivar groups and genetic structures.** Donut and pie charts indicate composition of chloroplast haplotypes and clusters identified in STRUCTURE, respectively. The green, orange, and blue in the donut charts indicate HAP3, HAP4, and HAP5, respectively. The colors for the pie charts are based on the results shown in Fig 2.

were found only among cultivars originated in China, not between cultivars from Japan and China. Most Chinese chestnut genotypes introduced into Japan were assumed to be from seeds from China or Korea [13]. Because seeds can be produced via outcrossing, their progeny would be genetically diverse. Clonal propagation would have been difficult compared with seed propagation because of graft incompatibility between Japanese and Chinese chestnut. In

**Table 5. Genetic characteristics of Chinese chestnut cultivar groups analyzed using 31 SSRs.**

| Group | Number of cultivars | $H_O$ | $H_E$ | AR | F |
|---|---|---|---|---|---|
| C_SJ | 27 | 0.544 | 0.599 | 4.35 | 0.085 |
| C_HE | 13 | 0.457 | 0.496 | 3.60 | 0.050 |
| C_SH | 25 | 0.570 | 0.603 | 4.40 | 0.066 |
| C_JI | 10 | 0.619 | 0.557 | 3.94 | −0.118 |
| C_AN | 9 | 0.642 | 0.567 | 4.00 | −0.137 |
| C_OR | 27 | 0.562 | 0.594 | 4.26 | 0.054 |

$H_O$ = Observed heterozygosity

$H_E$ = Expected heterozygosity

AR = Allelic richness

F = inbreeding coefficient

addition, some of the Chinese chestnut cultivars might not have been suited to Japanese climates, reducing the chance of introducing the Chinese chestnut cultivars by clonal propagation. The cultivars in C_SJ would have been selected either directly from seed introduced from China and Korea or from successive generations of hybridization among the introduced genotypes.

Genetic relationships among Chinese chestnut, Japanese chestnut, and Japanese–Chinese hybrid cultivars were clarified by Bayesian clustering analysis. Most of the Chinese and Japanese chestnut cultivars had simple genetic structure (Fig 1), indicating that those cultivars were pure Chinese chestnut or Japanese chestnut, respectively. The cultivar collections from Japan and China have been present in those nations for a long time, limiting the chances for interspecific hybridization. According to Isaki [13], introduction of nuts of Chinese chestnut into Japan began in the 20[th] century. On the other hand, Korean native chestnut had been considered as an intermediate between Japanese and Chinese chestnut on the basis of its morphological characteristics [43]. Consistent with that previous report, our study showed that the Korean cultivar 'Hamjung 3' (ID#62) was an admixture between Chinese and Japanese chestnut (Fig 1).

In addition to 'Hamjung 3', definitive introgressions of Chinese chestnut into Japanese chestnut and vice versa were identified in Japanese chestnut 'Obiwase' (ID#47) and Chinese chestnut 'Miyagawa 18' (ID#105). In a previous study, however, 'Obiwase' was presumed to have a genetic structure derived from a wild chestnut population distributed on Kyusyu Island; its structure was different from that of other native cultivars but it was still considered to be a pure Japanese chestnut cultivar [25]. Since this study did not include wild populations, this cultivar may have been miscategorized. Likewise, 'Miyagawa 18' carried both Chinese and Japanese chestnut genetic structure. The percentage of Japanese chestnut genetic structure was about 16%; thus, this cultivar would be a first backcross (BC1) or a second backcross (BC2), not an F1 hybrid. For Japanese–Chinese chestnut hybrid cultivars, both parentage and chloroplast haplotype analyses were conducted to determine the putative seed and parent cultivars. Out of 18 Japanese–Chinese cultivars, we were able to presume both parents for six cultivars and one parent for 10 cultivars (Table 4). Because Chinese chestnut cultivars were relatively uncommon in Japan and the hybridizations had been done relatively recently (within 100 years), many parent–offspring relationships were identified. The results of the analyses were compared with those of Isaki [13], who reported the parentages of some of the same cultivars. 'Shimaki 1', 'Shimaki 2', 'Shimaki 3', 'Shimaki 4', and 'Shimaki 5' (ID#77–81) were reported by Isaki [13] to be selected from seedlings derived from 'Ganne' and Chinese chestnut accessions. Here, the parentage of 'Shimaki 1', 'Shimaki 2', and 'Shimaki 5' was reconfirmed. Also, the parentage of 'Ikaba', listed in a plant variety protection database in Japan (http://www.hinshu2. maff.go.jp/en/en_top.html), matched our results. On the other hand, 'Hayashi 1', 'Hayashi 3', and 'Hayashi amaguri' were reported by Isaki [13] to be offspring of 'Kasashi 1', an F1 hybrid between 'Kasaharawase' (ID#43) and a Chinese chestnut accession; however, in the present study they were presumed to be offspring of 'Kasaharawase' or 'Kanotsume' (ID#24).

The Japanese–Chinese chestnut hybrid cultivars showed nut size intermediate (20.0 g) between those of Japanese (24.7 g) and Chinese chestnut cultivars (10.9 g). The average nut harvest date of the Japanese–Chinese chestnut hybrid cultivars was the similar to that of Japanese chestnut and earlier than that of Chinese chestnut (S3 Table). Thus, for Chinese chestnut breeding, Japanese chestnut cultivars have the potential to increase nut weight and shorten the time to harvest. On the other hand, introducing Chinese chestnut cultivars into Japanese chestnut breeding programs would be not an effective way to shorten the time to harvest or increase nut weight. Today, some Japanese–Chinese cultivars are highly valued by Japanese farmers and consumers because of the high nut quality and moderate nut pellicle peelability.

QTL analyses of interspecific backcross populations are necessary to identify interesting species-specific genes and accelerate breeding programs.

The origin of hybrid cultivar 'Riheiguri' (ID#75), one of the major cultivars in Japan, is an interesting example of interspecies hybridization giving rise to a desirable new cultivar. 'Riheiguri' was assumed to be an F1 hybrid between Japanese and Chinese chestnut since it had been developed in a Tsuchida orchard that contained both species [13]. Although we were unable to identify its parents by parentage analysis, Bayesian structure analysis revealed it had approximately equal membership in both the "red" and "light green" clusters, supporting the assumption that it was an F1 hybrid. In addition, 'Riheiguri' had HAP3, which originated in Jiangsu and Anhui. This cultivar might have been selected from seeds of other cultivars originated in these regions. It has relatively large nut size like Japanese chestnut (23.9 g; S3 Table), mealy texture like Chinese chestnut [13], and moderate pellicle peelability [44], which was likely inherited from Chinese chestnut. Because of its good nut quality, 'Riheiguri' and its relatives have been used in breeding programs, resulting in the release of new hybrid cultivars such as 'Shuhou' and 'Mikuri'. On the other hand, trials to release Japanese chestnut cultivars with the easy-peeling pellicle trait from Chinese chestnut have not yet succeeded [4].

We identified four chloroplast haplotypes (HAP3–HAP6) among the Chinese chestnut cultivars. HAP4 was mainly found in cultivars from Hebei, Shandong, and Japan, whereas HAP3 dominated in most groups except for C_HE. HAP5 and HAP6 were each identified in only one cultivar. At least 38 chloroplast haplotypes were identified from wild populations by Chen and Huang [19] and Liu et al. [17] using cpSSRs. However, only two haplotypes [19] and four haplotypes (this study) were identified from Chinese chestnut cultivar collections, suggesting that cultivars have limited genetic diversity compared to wild populations. Because the numbers and types of markers were quite similar in those studies and ours, it is reasonable to compare the results. On the other hand, it is possible that more chloroplast haplotypes would be found if we detected a larger number of polymorphisms by sequencing the whole chloroplast genome of several cultivars. The finding that cultivars showed less genetic diversity than wild populations corresponds to the suggestions of Mattioni et al. [45] and Ovesná et al. [20], i.e., that because traits and genes useful for chestnut cultivation were artificially selected, the domestication process would typically reduce genetic diversity.

Both chloroplast haplotype and Bayesian clustering analyses showed that the Chinese chestnut cultivars used in the present study could be divided into two groups (Fig 3): one originated in the Hebei region and the other originated in Jiangsu and Anhui. Most of the cultivars from Shandong had admixed genetic structure (Fig 2), whereas cultivars selected in other regions of China had various patterns of genetic structure. The cultivar groups that had admixed structure showed higher $H_E$ and AR, which is not unexpected because hybridization between cultivars from different clusters would increase genetic diversity. Since we had no information about cultivars selected in other regions of China, it was quite difficult for us to clarify the breeding history of these cultivars. Some cultivars from Shandong might have been selected from crosses between cultivars derived from Hebei and from Jiangsu or Anhui. Alternatively, cultivars might have been selected from wild chestnuts with an admixed genetic structure growing in Shandong. Although $H_E$ and AR were low in C_HE, C_JI, and C_AN, the values of $F$ were positive in C_HE but negative in C_JI and C_AN, indicating that artificial selection pressure was higher in the Hebei region. According to Kikuchi [46], nuts of cultivars from Hebei are small and sweet, whereas cultivars from central China have comparatively large nut size and low flavor, and cultivars from Shandong have intermediate characteristics. Consistent with this previous report, the average nut weight of Chinese chestnut cultivars that carried HAP3, the only haplotype found in the cultivars from Jiangsu and Anhui, was larger than those carried HAP4, the most common haplotype found in the cultivars from Hebei (Tables 3

and S3).Thus, breeders could use cultivars showing differences in genetic structure according to the objectives of their breeding programs.

The Chinese chestnut cultivars selected in Japan showed various patterns of genetic structure. More than half of those cultivars showed a "light green"-dominated structure presumed to have originated in Jiangsu or Anhui, while some cultivars showed admixed structure or "red"-dominated structure presumed to have originated in Hebei. The cultivars selected by Houji in Kochi prefecture ('Houji 354', 'Houji 445', 'Houji 350', 'Houji 446', and 'Houji 480'; ID#91–95) had high membership in the "light green" cluster, and the cultivars selected by Miyagawa in Yamanashi prefecture ('Miyagawa 100', 'Miyagawa 18', 'Miyagawa 84', and 'Miyagawa 85'; ID#104–107) had high membership in the "red" cluster. According to PCoA analysis, the cultivars selected by Houji were closest to those originated in Jiangsu or Anhui, while the cultivars selected by Miyagawa were closest to those originated in Hebei (S1 Fig), suggesting that Houji cultivars were selected from seeds originated in Jiangsu or Anhui and that Miyagawa cultivars were selected from seeds originated in Hebei. Geographical data support this hypothesis because Hebei is located in the northern part of China, and Hakushu in Yamanashi, where Miyagawa cultivars were selected, is located in a cold, high-altitude part of Japan. According to one account, Miyagawa cultivars were introduced from Songchong in North Korea, which is relatively close to Hebei [13]. The nut sizes of Miyagawa cultivars (which had genetic structure originated in Hebei) were 5.2–9.2 g, whereas those of cultivars that had genetic structure originated in Jiangsu or Anhui averaged around 12 g (S3 Table), corresponding to previous reports that cultivars in northern China have smaller nut size than those in central China [13]. In this study, we analyzed only Chinese chestnuts from Japan and from the seaside of China, not from western or southern China. Fortunately, chloroplast haplotype analysis clarified that the Chinese chestnut materials from Japan and China were similar (Fig 3), suggesting that the Chinese chestnut cultivars collected in this study were sufficient basic materials to classify Chinese chestnut cultivars in Japan. However, we are sure that there are many other genetic structures in Chinese chestnuts in other regions. Further analyses, including cultivars and wild populations from all over China, would clarify the detailed genetic relationships and domestication process of Chinese chestnut.

## Conclusions

We used SSRs to genotype chestnut cultivars preserved in both Japan and China and to determine the genetic structure of Chinese and Japanese chestnut cultivars. The synonym groups and putative parentages of some Chinese chestnut cultivars were identified here for the first time. Most of the Chinese and Japanese chestnut cultivars had a simple genetic structure corresponding to their respective species, whereas Japanese–Chinese hybrid cultivars had admixed structures. The Chinese chestnut cultivars could be divided into two groups: one that originated in Hebei and one that originated in Jiangsu and Anhui. The Chinese chestnut cultivars selected in Japan also carried the genetic structures originated in these two divergent regions, suggesting that their ancestral genotypes originated in those two groups. The information obtained in this study will be useful for population genetic studies for those species and for Japanese and Chinese chestnut breeding programs.

## Supporting information

**S1 Table. Names, accession numbers, and genotype information for the 230 cultivars used in this study.**
(PDF)

**S2 Table. List of the 31 SSR nuclear markers used in the present study.** The linkage group (LG) and position (pos) of each marker are based on an integrated map of a Kunimi × 709–34 population (Kx709CP; Nishio et al., 2018).
(XLSX)

**S3 Table Nut harvesting day and nut weight of the cultivars preserved at the NARO Genebank.**
(XLSX)

**S1 Fig. Principal coordinate analysis (PCoA) plot generated from genetic distance calculations among the 116 Chinese chestnut cultivars in GenAlEx software.** Cultivars were classified as predominantly part of the "red" cluster (ratio of "red" > 0.8), the "light green" cluster (ratio of "light green" > 0.8), or admixed (0.8 > ratio of "red" > 0.2; shown here in black).
(TIF)

## Acknowledgments

The authors thank Dr. Hiroyuki Iketani for his valuable suggestions and discussions.

## Author Contributions

**Conceptualization:** Eiich Inoue.

**Data curation:** Sogo Nishio, Shingo Terakami, Eiich Inoue.

**Formal analysis:** Sogo Nishio, Shingo Terakami.

**Funding acquisition:** Sogo Nishio.

**Investigation:** Sogo Nishio, Yutaka Sawamura.

**Resources:** Sogo Nishio, Shuan Ruan, Norio Takada, Yukie Takeuchi, Eiich Inoue.

**Supervision:** Toshihiro Saito, Eiich Inoue.

**Validation:** Toshihiro Saito, Eiich Inoue.

**Writing – original draft:** Sogo Nishio.

**Writing – review & editing:** Toshihiro Saito, Eiich Inoue.

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
