## [Decision Letter · Decision Letter 0]

22 Jan 2020

PONE-D-19-32891

Genetic evidence that Chinese chestnut cultivars in Japan are derived from two divergent genetic structures that originated in China

PLOS ONE

Dear Dr. Nishio,

Thank you for submitting your manuscript to PLOS ONE. After careful consideration, we feel that it has merit but does not fully meet PLOS ONE’s publication criteria as it currently stands. Therefore, we invite you to submit a revised version of the manuscript that addresses the points raised during the review process.

To revise the manuscript, please carefully examine the issues raised by the reviewers, especially the three points raised by the reviewer 2.

We would appreciate receiving your revised manuscript by Mar 07 2020 11:59PM. To enhance the reproducibility of your results, we recommend that if applicable you deposit your laboratory protocols in protocols.io, where a protocol can be assigned its own identifier (DOI) such that it can be cited independently in the future. For instructions see: http://journals.plos.org/plosone/s/submission-guidelines#loc-laboratory-protocols

We look forward to receiving your revised manuscript.

Kind regards,

Hidenori Sassa

Academic Editor

PLOS ONE

Journal Requirements:

Reviewers' comments:

Reviewer's Responses to Questions

**Comments to the Author**

1. Is the manuscript technically sound, and do the data support the conclusions?

Reviewer #1: Yes

Reviewer #2: Partly

2. Has the statistical analysis been performed appropriately and rigorously? 

Reviewer #1: Yes

Reviewer #2: Yes

3. Have the authors made all data underlying the findings in their manuscript fully available?

Reviewer #1: Yes

Reviewer #2: Yes

4. Is the manuscript presented in an intelligible fashion and written in standard English?

Reviewer #1: Yes

Reviewer #2: Yes

5. Review Comments to the Author

Reviewer #1: Overall it is well written manuscript. Clustering patterns and genetic structure were not surprising given that the study population comprised accessions from two different species. I have following suggestions to further investigate the genetic relationships between accessions from different species and/or regions.

1. Figure 1: The group C_SJ is clustered (100% green cluster) with the Chinese accessions. It’s not clear if all the Chinese chestnuts selected in Japan were originated from the first introduction of seeds. Could some accession haven been selected from successive generation of natural hybridisation with Japanese genepool?

2. Lin 357-366: The argument presented to support the origin of C_SJ group of accessions is not strong. I suggest authors to present neighbour-joining (NJ) and principal component analysis (PCA) of nSSR genotypes to further support their hypothesis.

3. Line 150: Authors should describe the steps for constructing chloroplast haplotypes. Some comments on the extent of linkage disequilibrium would be helpful in order to check the integrity of haplotypes.

4. Could a relatively lower chloroplast diversity (hence a simpler genetic structure) be a result of only 5 cpSSR used in this study?

5. It would be useful if authors could provide information on the distribution of 31 SSRs across different chromosomes. Perhaps using 31 SSRs would only detect genetic diversity of limited regions of the whole genome, and could lead to biased inferences about population genetics. Perhaps authors could discuss this point.

Reviewer #2: Authors tried to clarify the origin and genetic characteristics of Chinese chestnut cultivars in Japan by using 31 nuclear SSR markers and 5 cpDNA SSR markers in this research. They said results obtained by this research would be useful for both Japanese and Chinese chestnut breeding program.

However objectives of this research remain unclear, please consider following points,

1. What is the purpose (or significance) to reveal the origin of Chinese chestnut cultivars introduced into Japan? If authors would like to understand the origin of Japanese chestnut cultivars in relation to Chinese chestnut, materials used in this study were not enough. More Chinese chestnut cultivars in China should be involved.

2. Nuclear SSR markers were applied to reveal the genetic structure of Chinese and Japanese cultivars. But the resolution of SSR marker seems too high to apply for different species i.e., Castanea mollissima and C. crenata. Therefore only three clusters such as Chinese cultivar cluster, Japanese cultivar cluster and hybrid cluster were obtained. Is it possible to use other molecular markers such as the sequence of single nuclear gene or gene encoding chloroplast genome?

3. Authors say this research is useful for chestnut breeding program. But it is too general. Is there any information concerning agricultural traits for used cultivars? For example, is there any excellent or desirable trait inherited from putative parent to offspring?

4. This manuscript seems suitable to submit to more specific journal focused on the plant breeding or woody plants genetics.

6. PLOS authors have the option to publish the peer review history of their article (what does this mean?). If published, this will include your full peer review and any attached files.

Reviewer #1: No

Reviewer #2: No

---

## [Author Response · Author response to Decision Letter 0]

25 Feb 2020

Response to the Associate Editor

We thank the Editor and both Reviewers very much for taking time to review our manuscript and for your comments. We have revised our manuscript accordingly.

Responses to the comments from Reviewer 1

Comment

1. Figure 1: The group C_SJ is clustered (100% green cluster) with the Chinese accessions. It’s not clear if all the Chinese chestnuts selected in Japan were originated from the first introduction of seeds. Could some accession haven been selected from successive generation of natural hybridisation with Japanese genepool? 

Response

There were some cultivars in C_SJ that had originated from the first introduction of seeds into Japan: for example, ‘Houji’ and ‘Miyagawa’ had records of breeding history indicating that they were directly selected from seeds introduced from China into Japan. But others would have been selected from successive generations of the introduced genotypes, as this reviewer suggested. To clarify this point, we revised several sentences in the Discussion (Line 314, 327-329)

. 

Comment

2. Lin 357-366: T The argument presented to support the origin of C_SJ group of accessions is not strong. I suggest authors to present neighbour-joining (NJ) and principal component analysis (PCA) of nSSR genotypes to further support their hypothesis. 

Response

We performed principal coordinate analysis (PCoA; S1 Fig) and describe the results in the indicated paragraph (Line 419-422).

Comment

3. Line 150: Authors should describe the steps for constructing chloroplast haplotypes. Some comments on the extent of linkage disequilibrium would be helpful in order to check the integrity.

Response

According to this suggestion, we now describe in the Materials and Methods how we classified the haplotypes (Line 159-162). We also added the positions of the 5 cpSSRs in the complete Castanea mollissima chloroplast genome (KY951992) to S1 Table.

Comment

4. Could a relatively lower chloroplast diversity (hence a simpler genetic structure) be a result of only 5 cpSSR used in this study?

Response

Only two haplotypes (Chen et al., 2009) and four haplotypes (this study) were identified from Chinese chestnut cultivar collections because small numbers of cpSSRs (4 and 5, respectively) developed by Sebastiani et al. (2004) were used in those studies. Liu et al. (2013) also used a small number of cpSSRs (7) to clarify the genetic relationship between wild populations, in which they identified 38 chloroplast haplotypes. It is reasonable to compare the results of our study and of those previous studies because the numbers and types of markers were quite similar. On the other hand, it is possible that more chloroplast haplotypes would be found if we sequenced the whole chloroplast genomes of several cultivars and identified SNPs. We revised the text to clarify this point (Line 382-389).

Comment

5. It would be useful if authors could provide information on the distribution of 31 SSRs across different chromosomes. Perhaps using 31 SSRs would only detect genetic diversity of limited regions of the whole genome, and could lead to biased inferences about population genetics. Perhaps authors could discuss this point.

Response

The information on distribution of the 31 SSRs is provided in S2 Table and shows that the markers were mapped on different chromosome regions. The number of SSRs in this study was larger than that in most population genetic studies. Since we applied reliable SSRs sampled from among more than 500 developed in several previous studies, we have confidence in our results. Using a large number of SNPs from next-generation sequencing (NGS) might increase the amount of information; on the other hand, genotyping using NGS is not always precise because genomes contain much duplication.

Responses to Reviewer 2

Comment

1. What is the purpose (or significance) to reveal the origin of Chinese chestnut cultivars introduced into Japan? If authors would like to understand the origin of Japanese chestnut cultivars in relation to Chinese chestnut, materials used in this study were not enough. More Chinese chestnut cultivars in China should be involved.

Response

We revised several sentences in the Introduction to clarify the objective of this study (Line 52-60). Interspecific hybridization has been done by chestnut breeders all over the world to introduce disease resistance genes, an easy-peeling gene, and QTLs associated with nut quality into their breeding materials. Therefore, classification of cultivars in Japanese, Chinese, and their hybrids is quite important for breeders.

At the present time, materials from China are difficult to introduce into other countries because the Nagoya protocol (2017) prohibits exchange of cultivars. We examined 84 Chinese chestnut cultivars from 4 different provinces in China, that are moderate numbers and quite valuable for the first report of Chinese chestnut cultivar classification by SSRs. In addition, Chinese chestnut cultivars already in Japan have been preserved at the NARO (National Agriculture and Food Organization) Genebank (www.gene.affrc.go.jp) and can be readily used for research and breeding purposes all over the world. Thus, it is useful to clarify the genetic structure of these cultivars.

Comment

2. Nuclear SSR markers were applied to reveal the genetic structure of Chinese and Japanese cultivars. But the resolution of SSR marker seems too high to apply for different species i.e., Castanea mollissima and C. crenata. Therefore only three clusters such as Chinese cultivar cluster, Japanese cultivar cluster and hybrid cluster were obtained. Is it possible to use other molecular markers such as the sequence of single nuclear gene or gene encoding chloroplast genome?

Response

As the reviewer suggests, applying nuclear SSRs is not the best way to determine the genetic distance between different species. In this study, we applied SSRs only to identify synonym groups (first step) and to classify Japanese, Chinese, and Japanese–Chinese hybrid cultivars and determine the parentage of the hybrids (second step) not to determine genetic distances between cultivars of different species. The high level of SSR polymorphism was no problem for these analyses. We then performed additional classification within the group of Chinese chestnut cultivars (third step), which revealed two divergent genetic structures, i.e., Hebei origin and Jiangsu or Anhui origin. The values of Δ(K) were much higher at K = 2 than at K = 3 to K = 8 in Bayesian structure analysis, suggesting that there would be no more than 2 clusters in the Chinese cultivar collection.

Comment

3. Authors say this research is useful for chestnut breeding program. But it is too general. Is there any information concerning agricultural traits for used cultivars? For example, is there any excellent or desirable trait inherited from putative parent to offspring?

Response

To provide an example, we added the story of the hybrid cultivar Riheiguri, which is one of the major cultivars in Japan and has good nut quality and moderate peelability derived from Chinese chestnut (Line 362-375).

Comment

This manuscript seems suitable to submit to more specific journal focused on the plant breeding or woody plants genetics.

Response

Thank you for your comment. We considered our manuscript appropriate for submission because we have seen a number of similar population genetic studies reported in PLoS One, and we have attempted to follow the journal’s policies and guidelines.

---

## [Decision Letter · Decision Letter 1]

25 Mar 2020

PONE-D-19-32891R1

Genetic evidence that Chinese chestnut cultivars in Japan are derived from two divergent genetic structures that originated in China

PLOS ONE

Dear Dr. Nishio,

Thank you for submitting your manuscript to PLOS ONE. After careful consideration, we feel that it has merit but does not fully meet PLOS ONE’s publication criteria as it currently stands. Therefore, we invite you to submit a revised version of the manuscript that addresses the points raised during the review process.

I'd like to ask you to consider to add data of wild accessions and/or agronomic traits as suggested by the Reviewer 2.

We would appreciate receiving your revised manuscript by May 09 2020 11:59PM. To enhance the reproducibility of your results, we recommend that if applicable you deposit your laboratory protocols in protocols.io, where a protocol can be assigned its own identifier (DOI) such that it can be cited independently in the future. For instructions see: http://journals.plos.org/plosone/s/submission-guidelines#loc-laboratory-protocols

We look forward to receiving your revised manuscript.

Kind regards,

Hidenori Sassa

Academic Editor

PLOS ONE

Reviewers' comments:

Reviewer's Responses to Questions

**Comments to the Author**

1. If the authors have adequately addressed your comments raised in a previous round of review and you feel that this manuscript is now acceptable for publication, you may indicate that here to bypass the “Comments to the Author” section, enter your conflict of interest statement in the “Confidential to Editor” section, and submit your "Accept" recommendation.

Reviewer #1: All comments have been addressed

Reviewer #3: (No Response)

2. Is the manuscript technically sound, and do the data support the conclusions?

Reviewer #1: Yes

Reviewer #3: Partly

3. Has the statistical analysis been performed appropriately and rigorously? 

Reviewer #1: Yes

Reviewer #3: Yes

4. Have the authors made all data underlying the findings in their manuscript fully available?

Reviewer #1: Yes

Reviewer #3: Yes

5. Is the manuscript presented in an intelligible fashion and written in standard English?

Reviewer #1: Yes

Reviewer #3: Yes

6. Review Comments to the Author

Reviewer #1: Authors have addressed all my suggestions. However, I could not find Supporting Information S1 Fig. Can you please upload this information?

Reviewer #3: This manusript showed how cultivars are and how the relation among cultivars. It is important data for breeders. However, as for this level of journal (PLOSone), I feel wild accessions should be included or trait data. Particularly, author estimated the offspring and how many crosses had been done. Then, I am very interested in how their agronomical traits. And also in synonymous strains whether they share same kinds of traits.

Kinship is another issue. Even small number of SSRs, phylogenetic relationship is important data to consider breeding program. In addition, further research like GWAS, the preliminary data are required to conduct further research. In this meaning, authors can add these data and supposed to do. By the other hand, if not, authors are recommended to try other journals.

Minor comments

L220

Obiwase (#ID is required)

Some parts of cultivars are referred ID #, however, some are not. It is hard to follow the cultivar in Figure of STRUCTURE analysis to know how they are in data.

L240 "The sixth" is hard to get the meaning. One the sixth cultivars you mentioned?

Then, Just "Ikaba" is enough. If there are other Ikaba cultivars, please show ID#.

7. PLOS authors have the option to publish the peer review history of their article (what does this mean?). If published, this will include your full peer review and any attached files.

Reviewer #1: No

Reviewer #3: Yes: Ryuji Ishikawa

---

## [Author Response · Author response to Decision Letter 1]

26 Apr 2020

Response to the comment from Reviewer 1

Comment

However, I could not find Supporting Information S1 Fig.

Response

We added S1 Fig to the revised manuscript.

Responses to the comments from Reviewer 3

Comment

However, as for this level of journal (PLOSone), I feel wild accessions should be included or trait data. Particularly, author estimated the offspring and how many crosses had been done. Then, I am very interested in how their agronomical traits.

Response

We do not agree with Reviewer 3 on this point. We found a number of studies including only cultivars and accessions (i.e., not including wild accessions or phenotypic data) published in PLoS ONE within the past three years. Here are some examples:

Arnau, G., Bhattacharjee, R., Sheela, M. N., Chair, H., Malapa, R., Vincent Lebot, A. K., ... & Pavis, C. (2017). Understanding the genetic diversity and population structure of yam (Dioscorea alata L.) using microsatellite markers. PLoS ONE, 12(3), e0174150.

Manechini, J. R. V., da Costa, J. B., Pereira, B. T., Carlini-Garcia, L. A., Xavier, M. A., de Andrade Landell, M. G., & Pinto, L. R. (2018). Unraveling the genetic structure of Brazilian commercial sugarcane cultivars through microsatellite markers. PLoS ONE, 13(4), e0195623.

Urrestarazu, J., Errea, P., Miranda, C., Santesteban, L. G., & Pina, A. (2018). Genetic diversity of Spanish Prunus domestica L. germplasm reveals a complex genetic structure underlying. PLoS ONE, 13(4), e0195591.

Bernard, A., Barreneche, T., Lheureux, F., & Dirlewanger, E. (2018). Analysis of genetic diversity and structure in a worldwide walnut (Juglans regia L.) germplasm using SSR markers. PLoS ONE, 13(11), e0208021.

Zhu, S., Zhang, X., Liu, Q., Luo, T., Tang, Z., & Zhou, Y. (2018). The genetic diversity and relationships of cauliflower (Brassica oleracea var. botrytis) inbred lines assessed by using SSR markers. PLoS ONE, 13(12), e0208551.

Atnaf, M., Yao, N., Martina, K., Dagne, K., Wegary, D., & Tesfaye, K. (2017). Molecular genetic diversity and population structure of Ethiopian white lupin landraces: Implications for breeding and conservation. PLoS ONE, 12(11), e0188696.

Nevertheless, we added phenotypic data for nut harvest date and nut weight of cultivars preserved at the NARO Genebank in Japan. Such data were not available for the Chinese chestnut cultivars from China, for which we only had DNA samples.

Comment

And also in synonymous strains whether they share same kinds of traits.

Response

All of the synonym groups we identified were Chinese chestnut cultivars from China. As described above, only DNA samples were available for these cultivars, so we could not perform phenotypic analysis. However, in a previous study (Nishio et al. 2011), cultivars that had identical genotypes showed similar phenotypic traits.

Nishio S, Yamamoto T, Terakami S, Sawamura Y, Takada N, Saito T. Genetic diversity of Japanese chestnut cultivars assessed by SSR markers. Breeding Sci. 2011;61(2):109-20. doi: Doi 10.1270/Jsbbs.61.109. 

Comment

Kinship is another issue. Even small number of SSRs, phylogenetic relationship is important data to consider breeding program. In addition, further research like GWAS, the preliminary data are required to conduct further research. In this meaning, authors can add these data and supposed to do. By the other hand, if not, authors are recommended to try other journals.

Response

The results obtained using phylogenetic trees sometimes fluctuate depending on the parameters selected or the method of calculation. Instead of showing phylogenetic trees, we present the results of principal coordinate analysis (PCoA; S1 Fig) to clarify the genetic relationships between cultivars. The Bayesian structure and PCoA results will be useful for both chestnut breeding programs and further genetic studies including GWAS and population genetics.

Comment

Obiwase (#ID is required)

Some parts of cultivars are referred ID #, however, some are not. It is hard to follow the cultivar in Figure of STRUCTURE analysis to know how they are in data.

Response

In response to this suggestion, we now include both cultivar names and ID numbers in the text.

Comment

“The sixth" is hard to get the meaning. One the sixth cultivars you mentioned?

Then, Just "Ikaba" is enough. If there are other Ikaba cultivars, please show ID#. 

Response

In response to this suggestion, we combined and revised the sentences mentioned here.

---

## [Decision Letter · Decision Letter 2]

26 May 2020

PONE-D-19-32891R2

Genetic evidence that Chinese chestnut cultivars in Japan are derived from two divergent genetic structures that originated in China

PLOS ONE

Dear Dr. Nishio,

Thank you for submitting your manuscript to PLOS ONE. After careful consideration, we feel that it has merit but does not fully meet PLOS ONE’s publication criteria as it currently stands. Therefore, we invite you to submit a revised version of the manuscript that addresses the points raised during the review process.

The last point raised by the reviewer would be most important. Please carefully examine all the comments and revise the manuscript

We look forward to receiving your revised manuscript.

Kind regards,

Hidenori Sassa

Academic Editor

PLOS ONE

Reviewers' comments:

Reviewer's Responses to Questions

**Comments to the Author**

1. If the authors have adequately addressed your comments raised in a previous round of review and you feel that this manuscript is now acceptable for publication, you may indicate that here to bypass the “Comments to the Author” section, enter your conflict of interest statement in the “Confidential to Editor” section, and submit your "Accept" recommendation.

Reviewer #3: All comments have been addressed

2. Is the manuscript technically sound, and do the data support the conclusions?

Reviewer #3: Yes

3. Has the statistical analysis been performed appropriately and rigorously? 

Reviewer #3: Yes

4. Have the authors made all data underlying the findings in their manuscript fully available?

Reviewer #3: Yes

5. Is the manuscript presented in an intelligible fashion and written in standard English?

Reviewer #3: Yes

6. Review Comments to the Author

Reviewer #3: There some comments should be substituted or corrected;

Table 1

In cases of synonymous groups, authors omitted country name originated. Please use same notation.

199 Yanchang Beijing (China) C_OR (12)

Yanhong Beijing “no description of country here” C_OR (12)

Use alphabetical numeral letters in a case of less than 10.

253 we were able to infer both parents, 5 cultivars were F1 hybrids between Japanese and Chinese

254 chestnut and 1 cultivar, ‘Ikaba’ (ID#67), was presumed to be an offspring between Japanese–

353 18 Japanese–Chinese cultivars, we were able to presume both parents for 6 cultivars and one

354 parent for 10 cultivars (Table 4).

Five cultivars, a (or one ) cultivars

Table 4 does it require top bar to adjust format of PLOSone? Please confirm

Insert a space “),m”.

387 Table),mealy

387 Table), mealy

Is there a chance authors to misclassify synonymous or different types based on restricted number of SSRs. Because no other detail characteristics you can add to this data. Only a single evidence is not enough. You should refer the possible case in discussion section.

Authors mentioned Japanese case in the following reference but they are restricted genetic resources. In china, probably more numbers of resources and there might be similar but not the same landraces. At least need to reveal the possibility.

Nishio S, Yamamoto T, Terakami S, Sawamura Y, Takada N, Saito T. Genetic diversity of

Japanese chestnut cultivars assessed by SSR markers. Breeding Sci. 2011;61(2):109-20. doi: Doi

10.1270/Jsbbs.61.109.

7. PLOS authors have the option to publish the peer review history of their article (what does this mean?). If published, this will include your full peer review and any attached files.

Reviewer #3: No

---

## [Author Response · Author response to Decision Letter 2]

9 Jun 2020

Responses to the comments from the Editor

We thank the Editor and both Reviewers very much for taking time to review our manuscript and for your comments. We have revised our manuscript accordingly. During the revision, we found that there was a set of duplicated genotypes in the C_IO. Therefore, we revised all of the tables and figures. Because we only discarded one genotype, these changes had no effect on result and conclusion.

Comment

The last point raised by the reviewer would be most important. Please carefully examine all the comments and revise the manuscript.

Response

We take the last point is the possibility of the misclassification of the synonyms. We added the probability of identity, that is quite low (6.56× 10−35). It is unlikely to detect false synonyms with the 31 SSRs. We would like to show you the number of markers which can distinguish in the combination of all the two varieties (N_of_markers_distinguish.xlsx). Except for synonyms, cultivars can be distinguished with small numbers of markers. 

Responses to the comments from Reviewer 3

Comment

In cases of synonymous groups, authors omitted country name originated. Please use same notation.

199 Yanchang Beijing (China) C_OR (12)

Yanhong Beijing “no description of country here” C_OR (12)

Response

In response to this suggestion, we add the country names.

Comment

Use alphabetical numeral letters in a case of less than 10.

253 we were able to infer both parents, 5 cultivars were F1 hybrids between Japanese and Chinese

254 chestnut and 1 cultivar, ‘Ikaba’ (ID#67), was presumed to be an offspring between Japanese–

353 18 Japanese–Chinese cultivars, we were able to presume both parents for 6 cultivars and one

354 parent for 10 cultivars (Table 4).

Response

In response to this suggestion, we revised the sentences.

Comment

Is there a chance authors to misclassify synonymous or different types based on restricted number of SSRs. Because no other detail characteristics you can add to this data. Only a single evidence is not enough. You should refer the possible case in discussion section.

Authors mentioned Japanese case in the following reference but they are restricted genetic resources. In china, probably more numbers of resources and there might be similar but not the same landraces. At least need to reveal the possibility.

Response

We do not think there are misclassification of synonymous based on the 31 SSRs. We added the probability of identity, that is quite low (6.56× 10−35). It is unlikely to detect false synonyms with the 31 SSRs.

For fruit and nut species, it is typical that cultivars were spread by clonal propagation and the new names were given by local breeders and farmers. Because the cultivars in tree species have high heterozygosity and number of alleles in each locus are high, it is not difficult to distinguish cultivars using small numbers of SSRs. In most cases, the number of the SSRs to identify the synonyms were less than ours.

---

## [Decision Letter · Decision Letter 3]

15 Jun 2020

Genetic evidence that Chinese chestnut cultivars in Japan are derived from two divergent genetic structures that originated in China

PONE-D-19-32891R3

Dear Dr. Nishio,

We’re pleased to inform you that your manuscript has been judged scientifically suitable for publication and will be formally accepted for publication once it meets all outstanding technical requirements.

Kind regards,

Hidenori Sassa

Academic Editor

PLOS ONE

Additional Editor Comments (optional):

Reviewers' comments:

Reviewer's Responses to Questions

**Comments to the Author**

1. If the authors have adequately addressed your comments raised in a previous round of review and you feel that this manuscript is now acceptable for publication, you may indicate that here to bypass the “Comments to the Author” section, enter your conflict of interest statement in the “Confidential to Editor” section, and submit your "Accept" recommendation.

Reviewer #3: (No Response)

2. Is the manuscript technically sound, and do the data support the conclusions?

Reviewer #3: Yes

3. Has the statistical analysis been performed appropriately and rigorously? 

Reviewer #3: Yes

4. Have the authors made all data underlying the findings in their manuscript fully available?

Reviewer #3: Yes

5. Is the manuscript presented in an intelligible fashion and written in standard English?

Reviewer #3: Yes

6. Review Comments to the Author

Reviewer #3: (No Response)

7. PLOS authors have the option to publish the peer review history of their article (what does this mean?). If published, this will include your full peer review and any attached files.

Reviewer #3: Yes: Ryuji Ishikawa

---

## [Editor Report · Acceptance letter]

18 Jun 2020

PONE-D-19-32891R3 

Genetic evidence that Chinese chestnut cultivars in Japan are derived from two divergent genetic structures that originated in China 

Dear Dr. Nishio:

I'm pleased to inform you that your manuscript has been deemed suitable for publication in PLOS ONE. Congratulations! Your manuscript is now with our production department. 

Kind regards, 

on behalf of

Dr. Hidenori Sassa 

Academic Editor

PLOS ONE